# OPEN SOURCE BLOGGING WITH AUTOMUNGE

**Nicholas J. Teague** *
Automunge
Altamonte Springs, FL, USA
automunge@gmail.com

## ABSTRACT

The developers of the Automunge open source platform for tabular data preprocessing have taken a somewhat unorthodox approach to documentation and communications, making use of multimedia, blogging, tweets, jupyter notebooks, as well as music and photography in publication. This submission will offer an exhibited excerpt of such communication practices, featuring elements of multimedia videos with narration, accompanied with hand drawn slides and transcript, presented as both a brief introduction and extended walkthrough. We believe this form of presentation is a very accessible low cost option to communicate complex subject matter in a concise and accessible form. Further examples are also provided in the references.

## 1 INTRODUCTION

The Automunge open source python library platform for tabular data preprocessing [1] has adopted an open communications strategy dating back to the very start of development. We have a collection of blog posts [2] available on the Medium platform that have followed a regular publishing frequency throughout the approximately two years of development, generally with new essays published around every month. We've used these posts to document nearly every major design decision, experiment, and tutorial associated with the library. We have found that such a regular publishing schedule coinciding with development activities are mutually reinforcing, meaning beneficial in both directions. This approach has facilitated a rapid pace of iterative development and documentation that may otherwise have been slowed by pursuing more formal publishing channels.

In addition to Medium posts, we've also accumulated several conventions that we have found beneficial for a cohesive communications strategy, with code and tutorial notebooks shared on GitHub [3], and each software update elaborated on by tweets capturing screenshots of rollout notes [4]. The form of Medium posts are also somewhat fluid, with essays ranging from very informal blog posts with code demonstrations [5], architecture walkthroughs [6], academic papers [7, 8], and everything in between [9]. We have often incorporated elements of photography and music [10] to make at times dry subject matter more digestible to a wider audience.

## 2 EXHIBITS

Presented here will be two examples [11, 12] [Figures 1, 2] of explanatory blogging incorporating elements of multimedia presentation with slides and transcript. We took the approach of presenting the same information in two forms: an abbreviated adaption of about 6 minute video length for a brief introduction, and then an expanded version of the same presentation lasting about 33 minutes for those interested in a deeper dive.

We used as our primary tool for slide development an iPad with pencil and sketching app (particularly the GoodNotes app). We then recorded slide length narrations with Voice Memos that were pasted together with the images in the iMovie app on macOS. The videos were presented as an embedded YouTube video in a Medium post followed by static images of each slide coupled with the corresponding transcript of the narration.

---

*https://www.automunge.com

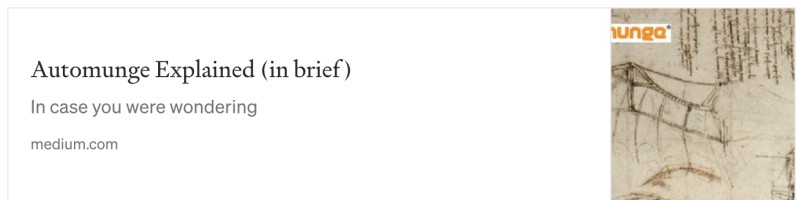

Figure 1: Automunge Explained (in brief)
https://medium.com/automunge/automunge-explained-in-brief-354c9b92aa1c

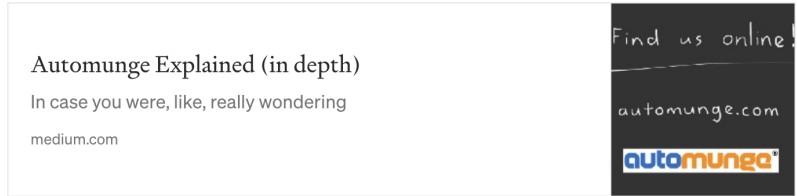

Figure 2: Automunge Explained (in depth)
https://medium.com/automunge/automunge-explained-in-depth-77ff777f12d7

We believe this form of presentation is a very accessible low cost option to communicate complex subject matter in a concise and accessible form suitable for a diverse selection of audiences.

## 3   DISCUSSIONS

- We found the video format with slides and narration particularly well suited for presenting the same material suitable for different audiences, as the video could be reviewed separately or in conjunction with the transcript and static slide images. Thus the same material could benefit virtual conferences for passive viewing as well as as serving as reference material for more active review. By presenting the same material in both a brief and extended duration we also were able to accommodate readers with varying interest.

- From a version control standpoint, we do not believe YouTube allows revisions to released videos, so there is some limitation for versioning, although a new video can always be uploaded with the link replaced if needed. We generally follow in our blogging that all material edits after the first day of publishing are recorded in the comments and with tweets for time stamps for purposes of version clarity.

- Since these videos are documenting an open source library, we believe the best bibliometrics suitable are corresponding activity like GitHub stars or PyPI download statistics.

- The interoperability for conversion (such as between blog posts and pdfs) are pretty fluid as the information content of the videos are fully redundant with the static slides and transcript. Note that the video itself can be embedded as a YouTube video in other pages independent of the Medium post if desired.

- We expect YouTube may be a dependable archive from a durability standpoint. The Medium platform, although less proven, has backup capabilities as an author's catalogue can at any time be downloaded and imported into other platforms (such as Wordpress).

- From an accessibility standpoint, we believe this form is very open to people of disability, as the same content is provided in video with audio and transcript, allowing comparable review by the blind or deaf for instance.

- Of course blogging has plenty of prior precedent. It is common for researchers to share blog posts corresponding to formal papers, as well as social commentary like twitter threads. Videos and slides have become a standard for poster presentations in online conferences.

- Some limitations of the form include lack of double-blind reviews. We believe that in this era of accelerating volume of research, there is a place for non-peer reviewed work, especially for emerging researchers as they work to build competence and expertise.

## 4 CONCLUSION

We hope that these exhibits may serve as an example of good practice for explanatory communications utilizing elements of multimedia with attached slides and transcripts. We also hope that this paper may serve as an introduction to a very useful library for tabular data preprocessing. For more details on Automunge there are several further resources linked in the References.

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
