# OpenReview forum: "Open Source Blogging with Automunge"
_ICLR.cc/2021/Workshop/Rethinking_ML_Papers/Exhibit_and_Workflow — Rethinking ML Papers - ICLR 2021 workshop Poster_

### Official Review · Reviewer_FG28 · 2021-03-28
**interesting ideas, but needs more structure and discussion of related work**

**Accessibility:**

Score of 1 (Weak): Submission does not commit to improving accessibility.

**Groundsforrejection:**

**Lack of clarity and structure in the work**
As it stands, the work seems more like a collection of various ideas, rather than an argument for a specific form that a ML paper could take. These ideas are interesting, such as using photography and music to explain ideas, but I am left wondering what precisely the proposed framework is. Unfortunately, there is too much work left to the reader to piece together how the work in Automunge translates to changes in how ML papers should be written.

I also wanted to see more explicit argumentation in favour of certain types of presentation. While there is one statement in the discussion about the accessibility of providing content in audio and transcript form, there is a missed opportunity to discuss the specific, possible benefits of using non-text media. I am not familiar with this literature, but it seems like some citations into the literature on how different media affect how people learn seem relevant.


**Lack of discussion of related literature**
Given that the ideas contained in the work and in the Automunge posts are not novel, I would have liked to see more discussion of how the work's approach compares with similar approaches that have been popular recently. These include:
- Videos and slides that have been the norm for online conferences
- Blog posts for academic papers
- Twitter explainer threads
How do these recent approaches contrast with the approach of Automunge? Are there transferable lessons to be drawn?


**Litreview:**

Score of 1 (Weak): The submission fails to acknowledge or cite previous work.

**Problemstatement:**

Score of 3 (Neutral): The submission states the problem, but either addresses a different one or fails to address the problem in a reasonably-significant manner (Note: as this is a new workshop, we’d like to be lenient with the word significant)

**Relevance:**

Score of 5 (Exceptional): Like (4) but does so with multiple themes of the workshop.

**Results:**

Score of 2 (Needs Improvement): Submission shows a poor level of clarity, novelty, coherency, and interactivity.

**Reviewerconfidence:**

I am fairly confident in my evaluation as it is based primarily on the work's lack of argumentation and relation to existing efforts to rethink ML papers.

**Reviewtext:**

This submission provides some examples of the way in which Automunge developers have documented their workflow through diverse means. After a brief presentation of how the documentation was done, there is some discussion on specific challenges and considerations the author encountered. Overall, the work implicitly argues for a multimedia form of presentation to accommodate audiences of varying backgrounds and abilities.

**Score:**

Reject: The reviewer believes the submission discusses important matters but it lacks clarity and design structure.

---

### Official Review · Reviewer_Lzou · 2021-03-30
**Open-Source Definitely, Double-Blind Not**

**Accessibility:**

Score of 3 (Neutral): Submission proposes methods to improve accessibility, but the level of intended accessibility is not well-articulated. Also, the limitations and exceptions are not stated.

**Groundsforrejection:**

Even if open-source blogging in ML is not a new concept -- the submission is interesting as it partially caters to what the workshop theme is and solves part of the problem statement. But close supervision of the workflow does bring out some caveats. The authors miss out on addressing a very important aspect of papers especially papers from peer-reviewed conferences i.e Double-Blind Review. It is expected that a system or method trying to redefine academic communication should consider solving this issue. As mentioned in the review, there are multiple existing open-source blogging methodologies on the internet, some of which are popular in the ML community such as https://distill.pub/ that follow a similar process of publishing. Authors should come up with building solutions over these existing frameworks that can incorporate the Double-Blind Review system and make academia free for all.

**Litreview:**

Score of 2 (Needs Improvement): The submission leaves out prominent examples of previous work in the area.

**Problemstatement:**

Score of 2 (Needs Improvement): The submission clearly has potential or credibility, but still fails to state the problem addressed clearly.

**Relevance:**

Score of 3 (Neutral): Attempt was clearly made to address a theme of the workshop, but it seems that the work was ‘retrofitted’ to match the theme of the workshop.

**Results:**

Score of 3 (Neutral): Submission is well designed and provides a good level of coherency/novelty/interactivity.

**Reviewerconfidence:**

4
The idea is simple and works to a certain extent. I am confident that my judgement on this submission is good and it only leaves some room for the author(s) to address the loopholes of their strategy.

**Reviewtext:**

The submission supports open-source blogging as a method for scientific communications through the authors' example of AutoMunge.

- The authors make a cogent case for interactive research through blog posts, version-controlling the pre-prints and updating changes on a Twitter thread. These are helpful alternatives over otherwise often-boring research PDFs which are difficult to version control.

-  Open-source blog posts need to be double-blinded in order to prevent academic gatekeeping and present no bias situations for fair publishing. Current implementations like distill.pub (I had expected seeing this documentation framework in the references) have been looked upon as great alternatives for research papers, however, they do not address this problem either.

# Judgement

The submission is interesting but solves some of the problems of academic paper publishing. It would be better if the authors propose some working solution to add double-blind reviews which work without the need for a PDF.

### Accessibility and inclusivity
Highly accessible, however detrimental when stereotypical gatekeeping is taken into account. Lack of double-blind systems in the blog post publishing process will hurt stereotypically and globally-south communities in ML research.

### Explainability and pedagogy
This submission scores high on work clarity and makes communication multi-modal.
Even if this is not something new (researchers already attach their papers with a blog post explaining their work), our community can but only hope that future iteration of the "PDF research paper" would have some elements of open-source blogging.

### Interpretability and visualization
This submission helps partially, the dissemination of complex knowledge in visual ways. Whiteboard-styled explanations seem helpful but some of these drawings and graphs are non-citeable (a part of a  video or one slide is never independently citable but only citeable in its entirety).

**Score:**

Reject: The reviewer believes the submission discusses important matters but it lacks clarity and design structure.

---

### Meta-Review · Program_Chairs · 2021-04-01

**Recommendation:** Accept
**Confidence:** 3

**Metareview:**

Pros: The exhibit discusses various different ways of presenting information about an Open Source library, Automunge. The core exhibit to explore would be those links from Figure 1, 2: they contain explainations about the library, in two videos that contain a mixture of formats.  There is a brief, overview video, as well as an in-depth video.  First, the videos are captioned.  Secondly, these videos have below them in the Medium posts a transcript with the slides from the video, to accomodate those who prefer text to video.

Cons: reviewers had difficulty identifying the key exhibit in this submission.

Reviewers were negative about this submission, but consulting the CfP for exhibits again, I felt that this exhibit was within what was solicited and double-blind, explicitly academic papers were not the only focus of the workshop.

---

### Decision · Program_Chairs · 2021-04-01

Accept (Poster)